# Challenges for a Sustainable Food Supply Chain: A Review on Food Losses and Waste

Annalisa De Boni [1],*, Giovanni Ottomano Palmisano [2],*, Maria De Angelis [1] and Fabio Minervini [1]

[1] Dipartimento di Scienze del Suolo, della Pianta e degli Alimenti, Università degli Studi di Bari Aldo Moro, Via Amendola 165/a, 70126 Bari, Italy
[2] Department of Economics, Università degli Studi di Foggia, Via Caggese 1, 71121 Foggia, Italy
* Correspondence: annalisa.deboni@uniba.it (A.D.B.); giovanni.ottomano@unifg.it (G.O.P.)

**Abstract:** To address global food security, new strategies are required in view of the challenges represented by Climate Change, depletion of natural resources and the need to not further compromise the ecosystems' quality and biodiversity. Food losses and waste (FLW) affect food security and nutrition, as well as the sustainability of food systems. Quantification of the adverse effects of FLW is a complex and multidimensional challenge requiring a wide-ranging approach, regarding the quantification of FLW as well as the related economic, environmental and social aspects. The evaluation of suitable corrective actions for managing FLW along the food supply chain requires a system of sound and shared benchmarks that seem still undefined. This review aims to provide an overview of the environmental, economic and social issues of FLW, which may support policy measures for prevention, reduction and valorization of food wastes within the food supply chain. In fact, detection of the hotspots and critical points allows to develop tailored policy measures that may improve the efficiency of the food supply chain and its sustainability, with an integrated approach involving all the main actors and considering the several production contexts.

**Keywords:** food losses; food waste; food byproducts; sustainable food systems; food security and nutrition; food quality

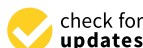



## 1. Introduction

In the last decades, the pressure on natural resources and ecosystems has strongly increased with the need for food for the growing world's population. Between 702 and 828 million people in the world faced hunger in 2021. Hunger affected 46 million more people in 2021 compared to 2020 and 150 million more people since 2019, before the COVID-19 pandemic. More than half of the people in the world affected by hunger in 2021 lived in Asia and more than one-third in Africa [1]. To address global food security, new strategies are required in compliance with the current economic and environmental constraints, and in view of the challenges represented by Climate Change, the depletion of natural resources and the need to not further compromise the ecosystems' quality and biodiversity [2]. The High Level Panel of Experts on Food Security and Nutrition [3] defined food losses as the "decrease, at all stages of the food chain prior to the consumer level, in mass, of food that was originally intended for human consumption, regardless of the cause" and food waste as "food appropriate for human consumption being discarded or left to spoil at consumer level–regardless of the cause". Both food losses and waste (FLW) generation have dramatically increased over the last few decades [4–6]; thus, their management and recovery strategies have been considered within the United Nations' Agenda for Sustainable Development [7], notably by the Sustainable Development Goal "Responsible consumption and production" (SDG 12). Within this Goal, the Target 12.3 (FLW reduction) seeks to halve per-capita global food waste at the retail and consumer levels and to reduce food losses along production and food supply chains (FSC), including postharvest losses [7]. Furthermore, FAO [8] identified the connections of SDG 12 and

Target 12.3 with almost all the other SDGs. In particular, they are strongly linked to the SDG 2 that deals with ending hunger and achieving food security and improved nutrition. Possible environmental effects fall under SDG 6 (sustainable water management), SDG 11 (sustainable cities and communities), SDG 13 (climate change), SDG 14 (marine resources) and SDG 15 (terrestrial ecosystems, forests, land and biodiversity). There are also knock-on effects on further SDGs: SDG 1 (ending poverty), SDG 8 (sustainable economic growth and decent employment) and SDG 10 (reducing inequalities). Moreover, progress on other SDGs, such as SDG 5 (gender equality), SDG 7 (affordable and clean energy), SDG 9 (infrastructure, industry and innovation) and SDG 17 (partnerships), could have beneficial impacts in terms of FLW reduction [9,10].

As a consequence, FLW impact the sustainability of food systems and their capacity to ensure global food security and nutrition in the long term [11]. FLW pose problems also for the ethics of wasting food in a world with increasing food insecurity in developed and developing countries [12]. Plant-based food is a key source of minerals and energy, providing almost 90% of the calories and 80% of protein requirements for daily human intake. Globally, about 210,000 M of hectares of agricultural land produces about 300,000 million tons (Mt) of vegetables [13]. Indeed, the amount of food produced on farmers' fields is much more than is necessary to feed humanity, but at the same time, losses of food between the farmer's fields and the dinner table are huge and aggravate malnutrition [3,5,14,15]. Therefore, reducing FLW is widely seen as a key strategy to reduce production impacts and increase the efficiency of the food system, improve food security and nutrition and contribute to environmental sustainability [8]. Additionally, the reduction of FLW could contribute to increase the income, not only for farmers, but also for processors, transporters, retailers and food service providers. In turn, a decrease of food prices, with a consequent increase of demand and consumption, could be achieved [16,17]. Although the interest of the scientific community is growing, there is still a gap of knowledge about the real extent of FLW, especially in terms of environmental, economic and social standpoints, taking into account the stages of food supply chain.

In this challenging context, the aim of this paper is to provide an overview of the environmental, economic and social issues of FLW. This study may be helpful in identifying FLW causes and supporting the development of policy measures enhancing the sustainability of FSC. This aim is pursued by performing an extensive narrative literature review, which enables to identify and summarize what has been previously published, avoiding duplications and seeking new study areas not yet addressed [18]. The narrative literature review allows putting many pieces of information together into a readable format, and it is a valid approach to present a broad perspective on a topic and to describe the history or development of a problem or its management. Moreover, the narrative literature review can serve to provoke thought, as it can be an excellent venue for presenting philosophical perspectives in a balanced manner and stimulating scholarly dialog amongst readers. For instance, readers can participate in this process by writing letters to the editor section of the journal and presenting their opinions and critical appraisal [19].

## 2. Quantification of FLW in the European Union (EU)

A homogenous framework for the quantification of FLW has not yet been defined and shared at European level [2,20,21]. Worldwide, it is estimated that about 1.3 billion tons of food is wasted annually [8,22,23], with a wide variability of the same figure at European level. According to estimates by Corrado and Sala [15], FLW annual per capita production ranges between 158 and 298 kg. As part of the FUSIONS project, analyzing the data on the amount and composition of waste in European countries, it has been estimated that 88 Mt of FLW is produced annually in the European Union (EU-28) and that about 20% of the total food manufactured is wasted [17]. The estimated per capita waste is about 173 kg/person and includes both edible and inedible food parts. This figure has been used as a benchmark for policies (for example, in the farm-to-fork strategy) [24]. Some researchers [2,20], who have analyzed the entire food chain through an analysis of mass

flow, reported much higher data: according to their estimates, an input of about 638 Mt of food raw materials should correspond to a production of about 129 Mt of "fresh" FLW.

Several studies have deepened the analysis in order to break down the waste stream by allocating the quantities of waste material at the different stages of production, distribution and consumption of food [14,25–28]. Waste has very different characteristics depending on the stage of the FSC in which it is generated [29]: in general, food losses (FL) generated in the upstream stages of the FSC (i.e., production, harvesting and processing) are poorly differentiated and contain a great quantity of raw waste of a limited number of types; waste from the downstream stages of the FSC (for example, distribution, retail and domestic consumption) contains smaller volumes of residues that are much more heterogeneous [30]. Scholars have generally agreed that farming and husbandry activities produce FL mainly at the postharvest stage, due to agricultural residues (e.g., roots and straw), unharvested crops and losses during harvest and spoilage during storage. This occurs mainly due to a lack of storage facilities, non-compliance with quality standard or required shape or appearance, farmer–buyer sales agreements failure and inefficiencies in transportation between farm and distribution [25]. Alexander et al. [26] estimated that globally a share of about 8.4% of primary crops harvested is lost. Other authors estimated losses in the USA from 2 to 23%, depending on the commodity, and about 9% in the UK [14,25,30–32]. In the definition and quantification of FL in the postharvest phase, it is essential to keep perishable and non-perishable foods separately [14]. In developed countries, waste is low for non-perishable crops such as cereals. For instance, losses are very low for barley, which can be as low as 0.07–2.81% under normal circumstances and mainly due to exogenous factors such as moisture, insects or rodents [14,25,30–32]. Oppositely, for perishable crops the waste is generally higher and mainly due to non-compliance with quality and size standards. These losses depend on the inefficiency of the handling systems during harvesting and packaging, cold storage and refrigerated transport facilities, scarce development of marketing systems dealing with "sub-standard" products and lack of integration among producers and marketers, which negatively affects logistics, supply management and delivery planning [25,31,32]. Table 1 summarizes the different authors' results regarding quantification (in kg per capita per year) of FLW at the different stages of the FSC.

**Table 1.** Amount of FLW (kg per capita per year) and share in each phase of the food supply chain.

| Total Amount (kg/y/Capita) | Share (%) at Each Phase of the Food Supply Chain | | | | | References |
|---|---|---|---|---|---|---|
| | Primary Production | Livestock Production | Processing and Manufacturing | Retail and Distribution | Consumption | |
| 180 | - | - | 39 | 5 | 56 | [33] |
| 275 | 47 | - | 12 | 7 | 34 | [5,22] |
| 289 | 43 | - | 12 | 5 | 40 | [34] |
| 173 | 11 | - | 19 | 5 | 65 | [17] |
| 179 | - | - | 39 | 5 | 56 | [35] |
| 290 | 9 | - | 21 | 12 | 58 | [36] |
| - | 8.4 | 12 | 36 | 19 | 33 | [26] |
| 257 | 25 | | 24 | 5 | 46 | [2] |

The different methodologies applied for data collection and evaluation lead to data variability. The most important references are the Accounting and Reporting Standard for Food and Waste Losses [37], developed jointly by different institutions (e.g., WRI, FAO, WRAP, UNEP and WDCSD) [15], and the FUSIONS quantification manual [38] and European Union Directive 2008/98/EC [39]. It should be noted that the numerous studies and projects on FLW in the EU [2,5,17,26,31,33,34,36,40] refer to different definitions, methodologies and objectives, system boundaries and databases. Moreover, data are related to a variable number of countries with great difference in production processes, also providing different results in terms of amount of waste. The highest number of waste (33–65%) is related to consumption behaviors, and it is influenced by socio-economic consumers' features (gender, age, income, education, etc.), frequency and planning of shopping and food preparation, awareness and sensitivity to environmental issues [4,40,41] (Table 1). This share could be reduced by applying targeted prevention strategies turned to

consumers [42]. A share of waste between 12 and 39% comes from industrial processing and may arise from each food supply chain stage (Table 1). The high amount of waste in processing is mainly due to the spoilage of commodities during transport and storage and the scrap arising from washing, peeling, cutting and cooking operations. The amount of waste may further increase if there are accidental process interruptions, contaminations, inappropriate packaging or crops selection [25,27,28]. The main causes of losses in the distribution phase, ranging from 5% to 19%, are inefficient means of transport, unsuitable packaging and planning errors in the purchase and reception of goods (Table 1) [43–45]. The production of waste in the sales phase, both at wholesale and retail, is strongly related to the ability to predict demand, to program supply, to the management of stocks and inventories and to the management policies of unsold food. Food labeling may play a significant role because the indication "best before", "sell by" or "use by" may be easily misunderstood by consumers. To reduce the waste at retail level, an increasing number of food retailer and large-scale distribution firms cooperate with charitable organizations for food redistribution (e.g., food banks) or sell close-to-expiry products at discounted prices.

Caldeira et al. [2] estimated the contribution of FLW from each food group as well as the amount of "EU available" food commodities (calculated as production, plus import, minus exports of primary commodities, minus non-food uses). Figure 1 shows that contribution on a plane of Cartesian axes wherein the horizontal axis reports the food groups, while the vertical axis reports the FLW and food commodity availability as a percentage of their own total amount.

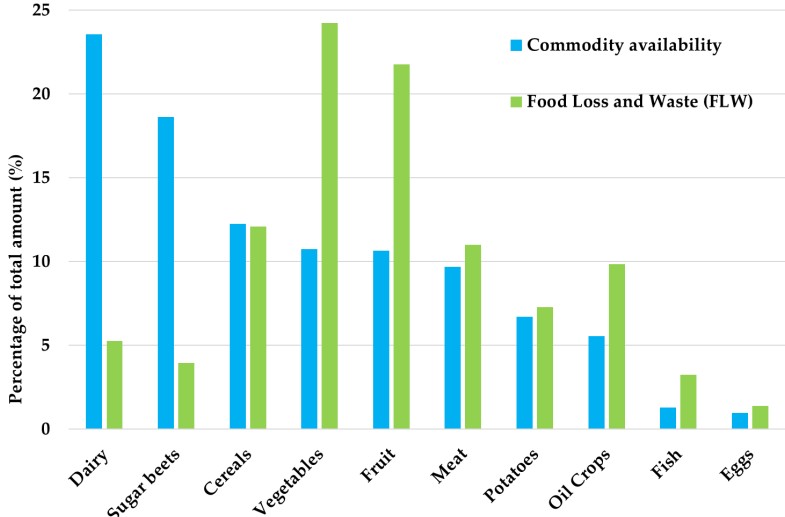

**Figure 1.** Contribution of FLW (green columns) and "EU available" food commodities (light blue columns). Data from Ref. [2].

According to the authors, the largest amount of FLW is related to vegetables (24% of total losses and waste) and fruits (22%), followed by cereals (12%), oil crops (10%), potatoes (7.3%) and sugar beets (4%). On the other hand, animal origin food, such as meat (11%), dairy (5.3%), fish (3.3%) and eggs (1.4%), generate the lowest quantities of FLW. Despite the growing interest of the scientific community, gaps still remain in the availability of data on the contribution of individual products or product groups to the total waste. The highest availability of information concerns the products that generate most of waste (cereals, fruits and vegetables) and products of animal origin [33], probably due to the increasing concerns about their environmental impact. The percentages of FLW have been reported as waste loss coefficients (in percentage) only by few authors [5,12,16,34]. Table 2 summarizes the coefficients of FLW per food group and stages of FSC [2]. Again, the high variability in the coefficients depends on differences regarding countries' supplying data, quantification data methodologies, FSC stages and the season during which data were attained and estimates were performed.

**Table 2.** Ranges of FLW coefficients (%) per food group and food supply chain stage.

| Food Supply Chain Stage | Food Groups | | | | | | | | References |
|---|---|---|---|---|---|---|---|---|---|
| | Fruit and Vegetables | Cereals | Meat | Oil Crops | Roots and Tubers | Dairy | Fish | Eggs | |
| Primary production | 18.0–20.0 | 1.5–4.3 | 0.8–3.2 | 2.5–10.0 | 2.8–20.0 | 0.3–3.5 | 9.4 | 4.0–4.8 | [12,20,35] |
| Storage and Handling | 5.0–7.3 | 3.9–4.0 | 0.7 | 1.0–1.2 | 7.6–9.0 | 0.5–1.7 | 0.5–7.9 | 1.9 | [20,35] |
| Processing and manufacturing | 2.0–6.4 | 3.2–10.5 | 4.7–5.0 | 5.0–28.2 | 4.9–15.0 | 0.7–1.2 | 6.0 | 0.5–1.6 | [12,20,35] |
| Retail and distribution | 1.2–10.0 | 2.0–3.0 | 2.8–4.0 | 0.7–7.0 | 0.3–1.0 | 0.3–0.8 | 9.0 | 1.6–2.0 | [12,20,35] |
| Consumption | 17.9–26.2 | 13.0–27.0 | 11.0–14.6 | 4.0–5.0 | 13.3–25.5 | 9.8–15.0 | 8.0–22.6 | 2.0 | [12,20,34,35] |

The analysis of the scientific literature shows a gap of knowledge regarding the data useful for quantifying the contribution of each phase of the supply chain (for example, primary phase, transformation, sale and consumption) to the total quantities of FLW. This step represents the starting point for planning, evaluating and identifying well-founded waste prevention actions.

## 3. Economic Evaluation and Costs of FLW

To the best of our knowledge, scientific literature has been more focused on FLW quantification, prevention, reduction and environmental impacts, and has devoted less attention to the economic assessment of FLW and food byproducts and wastes (FBPW). According to the Directive 2008/98/EC on waste [39], a byproduct is a substance resulting from a production process, which can be further used after standard industrial processing, excluding adverse environmental or human health impacts.

However, there is scientific evidence for the strong relations between economic performances, the behavior of individual actors in the FSC (i.e., consumers, producers, etc.) and the generation of losses and waste [30]. FAO explains in detail that the causes of FLW widely vary along the FSC and can be directly or indirectly linked to a specific point of loss [36]. The causes are generally context-specific, including inadequate collection times, climatic conditions, practices applied at the time of collection and handling, difficulties in marketing products and deficiency of economic incentives to prevent losses [37]. During transport, good physical infrastructure and efficient commercial logistics are vital to prevent further FL [38]. The estimation of the value of wasted food is mainly carried out following two approaches: (i) its cost of production and (ii) its market price [33]. Regarding the production phase, the estimated value of FLW and byproducts at the producers is about 75 billion euros, about 60 billion euros in the phases of postharvesting, processing and distribution, while the consumption stage generates a waste value of about 70 billion euros [2,39]. According to the results of the FUSIONS project, the costs related to FLW for EU-28 in 2012 were estimated at around 143 billion euros [8]. The consumption phase is responsible for most of this cost: domestic waste has been estimated at a value of about 98 billion euros, and postcollection, processing and distribution operations generate losses of about 43 billion euros, while the losses of the primary phase have been estimated of approximately 1.8 billion euros.

Apart from an overall economic quantification of the waste value, it is crucial to evaluate how the overall economic losses connected to FLW are shared among the food chain actors in relation to their market or purchasing power, and to consider the costs of FLW reduction [40]. Analyses that quantify the management costs of food recovery or its profitability are still poor [23,41]. Firms often underestimate the cost of waste, although a proper waste and byproducts management can be paramount in improving profits and the evidence and communication of the economic benefits, which may enhance the sustainability of FLW reduction policies. Moreover, evidence about the efficacy of actions to reduce FLW and improve economic sustainability are still scarce and do not guide key actors in obtaining economic incentives within a farm-to-fork virtuous approach [35]. Some

authors investigated the economics assessment and gains from adopting loss-reducing technologies [42,43]. In particular, the economic importance of losses is directly linked to their management along the FSC and to the resources devoted to loss-abating actions [42,44]. The economic value of FLW depends on volumes of products, on their position inside the market place, on the collection and transport costs and on the period of assessment [45]. Several authors [42,44,45] agreed that the costs of loss reduction actions should be taken into account in the economic assessment of food losses. With a view to a cost–benefit analysis aimed at quantifying the economic impact of management actions, it is important to consider all actors at different levels of the value chain involved in actions that aim to eliminate or reduce FL.

An effective waste management may have significant effects on profitability for all food chain operators [46], and this virtuous management can be implemented through different actions. First, an effective management of material and energy inputs and outputs of production processes can not only reduce production costs but also limit environmental impacts [46]. Second, an effective process of recovery of materials otherwise thrown away in the FLW process is feasible for waste management. Third, the application of digital technologies for smart agriculture (e.g., Internet of Things, blockchain, artificial intelligence, etc.) can prevent or reduce FLW by shortening harvesting and delivery times to the market, as well as improve the information exchange between producers and suppliers helping to schedule all the production and transportation processes [47,48]. An appropriate recovery, recycling and valorization of FLW involving their use to extract high-value compounds suitable for food, pharmaceuticals and cosmetics sectors may be obtained and provide natural antioxidants, preservatives and dietary supplements [4]. Another economic improvement that could be gained through the recovery of valuable components of FW is the reuse of wastewater. For example, olive mill wastewater could be valorized as a source of bioactive phenols and pectin [49,50].

## 4. Environmental Issues

FLW have a significant environmental impact due to the consumption of natural resources such as land, water, nutrients and energy. In fact, they not only boost the cost of food production but also have negative effects on the environment, namely biodiversity and nutrient losses [47,48]. For these reasons, estimating the environmental and economic costs related to FLW appears to be an important aspect to inform policy and design alternatives and sustainable strategies for food production and consumption [5]. A reduction in FLW is acknowledged to lead to positive environmental effects, but these consequences are highly dependent on the amount of avoided loss and waste, the stage in the supply chain where the savings are created and the saving costs achieved [6,49]. Life cycle analysis (LCA) is the main approach for the assessment of the environmental impact of FLW, applied with top-down approaches (data from input–output tables) or bottom-up, using databases from specific more detailed inquiries. Several scholars applied a bottom-up LCA approach to calculate the global warming potential (GWP) of waste, from databases of national statistics and/or direct consumption survey results [51,52]. Kummu et al. [51] applied a top-down approach to global statistics from FAO and estimated, at the country level, losses and wastes of food crops in terms of resources used for their production. For Europe, it has been estimated that 31% of the freshwater, 24% of the cultivated land and about a quarter of the fertilizers used to produce food flow into FL [53]. The impact of agricultural land use is related to various effects of cultivation: soil degradation, loss of landscape features and loss of biodiversity, etc. Cereals contribute to the overall impact expressed in terms of "land use" for about 45% of the cultivated area, legumes and oilseeds account for 30%, fruits and vegetables for 19% and, lastly, roots and tubers account for 6%.

The total greenhouse gas (GHG) emissions of food over its life cycle, expressed in kilograms of $CO_2$ equivalents (eq), was defined by FAO in 2013 [52]. GHG emissions of the agricultural phase include the emissions from soils and livestock (such as $CH_4$ and $N_2O$). According to FAO results, FLW global impacts are about 3300 Mt $CO_2$ eq of GHG,

240,000 m$^3$ of irrigation water wasted and 1.4 billion hectares needlessly cultivated. The total carbon footprint of Europe is about 495 Mt of $CO_2$ eq (700–900 kg of $CO_2$ eq per capita and per year) with a contribution of 15% to global greenhouse gas emissions [3]. It is possible to estimate the carbon footprint through the LCA, quantifying the contribution of each stage of the FSC as GHG emissions. Notarnicola et al. [53] underlined that the impacts associated with FLW are distributed in the impacts related to the different life cycle stages of the supply chain. For food systems, GHG emissions from primary production (agriculture, fishery, postharvest and handling) are always the major contributors to the carbon footprint. According to FAO [3] estimates, the contribution of FLW in the EU to the total production of GHG consists of 18% from the primary production phase, 18% from handling and postharvest storage, 15% from processing and production, 14% from retail and distribution and finally 35% from consumption. Cereals, meat and vegetables FLW generate more than 60% of the global carbon footprint [3]. These wastes each cause an emission of about 100 Mt $CO_2$ eq. Lost and wasted roots and fruits give a contribution of about 30 Mt $CO_2$ eq each, milk of 40 Mt $CO_2$, while oil crops and pulses, fish and seafood provide an emission of about 20 Mt $CO_2$ eq each. The overall climate impact of cereals is mainly due to the production and use of nitrogen fertilizer and fuels consumption for farming operations. Moreover, yield level strongly affects the emission of different cereal species.

Meat and dairy products' waste impacts are linked to GHG emissions from animals. For pigs and poultry (monogastric), the main component of emission is $N_2O$ from feed production due to mineral fertilizers production, followed by energy used for animal housing [54,55]. Manure management contributes to GHGs to a lesser extent. Ruminants rearing generates great CH4 emissions due to enteric fermentation and, in smaller measure, manure management [54–56]. For ruminants, feed production is the second source of emissions because of fertilizers and fuels employed in pasture and fodder production. Fruits and vegetables production is associated with low GHG emissions, mainly due to fertilizers and mechanization needs, which are proportional to the yield level, and farming techniques (e.g., heated greenhouses generate higher emission). Pulses and legumes, produced by nitrogen-fixing plants with modest nutritional needs, generate very low emissions, similarly to roots and tubers, due to their high yields that reduce GHG emissions per kg of product [54,55]. Scholars and policymakers devote attention also to the assessment of nitrogen and phosphorus emissions due to the well-known negative effects on the loss of nitrogen and phosphorus from agriculture and herds. Grizzetti et al. [57] estimated that 12% of the nitrogen pollution from agriculture in the EU is related to loss and wasted products. A further cause of concern is that nitrogen pollution from agriculture is among the causes of the eutrophication of European coastal waters [57,58], resulting in a threat to drinking water located in agricultural areas. Regarding the strategies for the reduction of the environmental impacts of FLW, recent research studies have shown that the application of digital technologies for smart agriculture can be crucial for achieving environmental sustainability of the food supply chain itself. These technologies may contribute to lowering both the carbon footprint along the entire chain and waste production [47,59,60].

## 5. Social Implications

The quantity, impacts and possibilities to limit waste are strongly related to a set of social factors, arising from mutual interactions between subjective and contextual elements [61]. Among the subjective factors, some authors [25] included the errors of farmers and food processors in the management of production, trade and logistics processes. small companies in particular may overestimate both the food demand and production need, due to inadequate facilities or skills and technical difficulties. Unforeseen price fluctuations and inefficient technical capacities can make harvesting uneconomical and cause field losses. Finally, the inadequacy of the processes of storage, packaging and processing can adversely affect both the agricultural phase and the phases after harvesting. Awareness and understanding of FLW, concerns about financial loss and the environmental and social

implications of FW are the main socio-demographic factors affecting food habits, household food-related practices and routines [27,62–64]. National and local policies implementing strategies for the collection and recycling of FLW are the external factors, playing a relevant role in limiting waste generation, preventing landfilling and incineration. These strategies trigger a circular economy approach, boosting the use of functional substances that can be recovered from waste [64,65] and must be adapted to the waste origin, categories and supply stages, considering the possibility of interdisciplinary collaboration [66]. These policies are generally carried out through the taxation of landfills or incineration, the payment of economic incentives to companies that reduce the production of FW, the implementation of efficient systems for the collection of municipal waste, the setup of information and public awareness campaigns. Furthermore, recent studies highlighted that the employment of digital technologies for smart agriculture may address certain social aspects of FLW, such as improving transparency and safety in the FSC to prevent disposal of edible products or loss through contamination tracing [48,67,68].

The opportunities related to the recovery of high-value compounds from FLW and FBPW depend on the availability of appropriate techniques for the waste and byproducts collection, management and extraction in compliance with ensuring traceability and safety and sustainability of high-value-added compounds, according to the European regulatory framework [23,39,69]. In the current scenario, this is still a challenging task for the researcher and entrepreneurs.

Many emerging technologies based on bioprocessing with selected microorganisms, fluids at high pressures, membrane filtration, enzyme extraction, microwaves, ultrasounds, high hydrostatic pressures and pulsed electric fields have been taken into account and are currently under study [70–72]. Table 3 reports some examples of recent studies dealing with the main bioactive compounds obtainable from widely available FBPW.

**Table 3.** Examples of the main bioactive compounds obtainable from widely available FBPW sources.

| Food Groups | FBPW Sources | Bioactive Compounds | References |
|---|---|---|---|
| Fruit and vegetables | Apple pomace | Fibers | [22] |
| | Apple seeds | Polyphenols | [73] |
| | Apple pomace | Pectine | [74] |
| | Tomato pomace | Polyphenols | [75] |
| | Tomato pomace | Lycopene | [76] |
| | Orange peels | Pectine | [74,77] |
| Vine crops | Grape pomace (white wine) | Fibers | [70] |
| | Grape pomace (red wine) | Fibers | [70] |
| | Grape pomace (red wine) | Polyphenols | [70] |
| | Grape pomace (white wine) | Polyphenols | [70] |
| | Grape pomace (white wine) | Fibers | [70] |
| Cereals | Brewers' spent grain | Proteins | [78] |
| | Brewers' spent grain | Fibers | [79] |
| | Brewers' spent grain | Fibers | [80,81] |
| | Brewers' spent grain | Polyphenols | [82] |
| | Maize bran and germ | Fermented byproducts | [71] |
| Meat | Poultry byproducts | Amino acids | [83] |
| | Poultry byproducts | PUFA | [83] |
| | Pig bones | Collagen peptides | [84] |
| Oil crops | Olive pomace | Polyphenol | [85] |
| Dairy | Cheese whey | Sophorolipids | [86] |
| | Cheese whey | Dairy starter cultures | [87] |
| | Cheese whey | Bacteriocins | [88] |
| Fish | Shrimp and different parts of crab | Chitin | [89] |
| | Shrimp and different parts of crab | Carotenoid pigments | [89] |
| | Fish waste | Protein hydrolysates | [90,91] |

The implementation of virtuous practices for lowering, collecting and recovering FBPW may contribute to the sustainable development of rural, coastal and industrialized

areas, notwithstanding the social acceptability results strongly influenced by the level of information and knowledge of involved firms and citizens [92]. Moreover, innovative supply chains recovering and up-cycling FBPW can create job opportunities, especially for small and medium-sized enterprises, favoring virtuous paths for local development [93]. However, it should be noted that consumers' knowledge and awareness about the nutritional quality and environmental performance of foods enriched with high-value compounds obtained from FBPW are still scarce. In fact, the willingness of consumers to accept these new foods is still low; thus, actions to enhance appreciation are required [94,95].

## 6. Conclusions and Future Prospects

Public authorities, institutions and non-governmental organizations at different territorial levels show an increasing attention and commitment to achieve food security goals and efficient and sustainable food systems. A deeper knowledge of the FSC can allow detecting the hotspots and critical points related to FLW, allowing to develop tailored policies and strategies to improve sustainability. In addition, the quantification of the adverse effects of FLW and the evaluation of suitable corrective actions require a system of sound and shared benchmarks that seems still undefined. The number of recent studies bears witness to the scientific community's interest and attention to issues related to waste generation, which is unquestionably a complex and multidimensional challenge. This issue requires a wide-ranging approach, addressing not only the economic, environmental and social aspects of sustainability, but also the involvement of all actors of the FSC (i.e., farmers, processors, distributors, sellers and consumers) from production to consumption.

The need to ensure food security and nutrition and the sustainability of food systems, in compliance with the SDGs, brings the urgency of a global commitment for the management and reduction of FLW. In order to make progress towards SDGs related to FLW reduction and to address the global food demand, the findings from this review suggest that the following four actions could be applied to the FSC: (1) to quantify how much food is lost and wasted, as well as where and why; (2) to explain and communicate the benefits of reducing FLW to the FSC actors; (3) to suggest strategies and tools for FLW reduction, optimizing the efficiency along the FSC; (4) to develop strategies to properly collect FBPW, recovering high-value compounds, ensuring their traceability, safety and sustainability.

In particular, the insights of this study may support the efforts of policymakers and government institutions, especially at the local level, in defining policy measures that have to be designed and planned carefully considering the measurements of losses and waste, the supply chain where they occur and the critical loss points. Therefore, tailored policy measures should be structured at three different levels.

Firstly, measures aimed at preventing waste generation and ensuring food security have to be strengthened in order to reduce inefficiencies in the food systems. For instance, these measures could improve: the management of edible and inedible parts of food at the primary production level and the efficiency of storage, packaging and processing by adopting digital technologies for smart agriculture; the reduction of waste at the retail level by the better management of "close-to-expiry" products and food redistribution, as well as by stimulating the consumers' awareness about food waste management through food and nutrition knowledge campaigns.

Secondly, specific policy measures could support the implementation of FW collection and treatment infrastructures and services, which should be designed in relation to the quantity and quality of food waste to be managed (e.g., recovery of high-value-added compounds, anaerobic digestion, composting, etc.). In this step, the quantification of environmental impacts, costs and benefits of FW reduction will be crucial to justify the targeted allocation of financial resources.

Finally, weighing the costs and benefits of FW reduction represents a guidance for designing policies in the view of the valorization of waste as energy/material (e.g., using FW to produce high-value compounds or biofuels). In this regard, the overview raised from this study may improve the awareness of the origin and composition of FW flows

within the FSC and to what extent these flows may vary over time. This may support the development of specific policy measures aimed at facilitating the implementation of innovative sustainable processes for bioactive molecules and functional ingredients recovery. These processes contribute to the efficient management of waste, meanwhile taking advantage of an inexpensive source of valuable compounds with recognized positive effects on human health.

As practical implications, these possible policy measures must necessarily be based on integrated food chain approaches taking into account the strong link between FLW reduction and food security, within the horizontal and vertical cooperation among the FSC actors and institutions. Indeed, effective measures for preventing and reducing FLW have to involve all the main actors and institutions in sharing concrete strategies and tools. However, it is useful to identify critical loss and waste points along the different supply chains as key steps for implementing these measures, which should be based consequently on the peculiarities of the several production contexts.

**Author Contributions:** Conceptualization, A.D.B. and F.M.; methodology, A.D.B. and F.M.; formal analysis, A.D.B., F.M. and G.O.P.; investigation, A.D.B., F.M. and G.O.P.; writing—original draft preparation, A.D.B. and F.M.; writing—review and editing, A.D.B., F.M. and G.O.P.; supervision, M.D.A.; funding acquisition, M.D.A. All authors have read and agreed to the published version of the manuscript.

**Funding:** This research was funded by the project SYSTEMIC "an integrated approach to the challenge of sustainable food systems: adaptive and mitigatory strategies to address climate change and malnutrition", Knowledge Hub on Nutrition and Food Security, which has received funding from national research funding parties in Belgium (FWO), France (INRA), Germany (BLE), Italy (MIPAAF), Latvia (IZM), Norway (RCN), Portugal (FCT) and Spain (AEI) in a joint action of JPI HDHL, JPI-OCEANS and FACCE-JPI launched in 2019 under the ERA-NET ERAHDHL (n° 696295).

**Institutional Review Board Statement:** Not applicable.

**Informed Consent Statement:** Not applicable.

**Data Availability Statement:** Not applicable.

**Conflicts of Interest:** The authors declare no conflict of interest.

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
