# Peer review of "Challenges for a Sustainable Food Supply Chain: A Review on Food Losses and Waste"

_sustainability, doi:10.3390/su142416764_

Round 1

Reviewer 1 Report

After reviewing this paper, I concluded that it lacks basic ingredients that are required for the acceptance of any review article. The authors have just summarized the previous studies without adding anything new to the prior literature. In my opinion, this could be a summary of previous studies but not an article in its current form. I couldn't even find the study's objectives, and the abstract exists in a different world than the rest of the manuscript. The authors aim to develop a framework for the environmental, economic, and social issues of FLW that may support strategies (as mentioned in the abstract), but the manuscript does not even discuss this framework in the later work. Therefore, with regret, I would request the editor to reject the manuscript this time.

Author Response

Reviewer #1

Comment 1a. After reviewing this paper, I concluded that it lacks basic ingredients that are required for the acceptance of any review article. The authors have just summarized the previous studies without adding anything new to the prior literature. In my opinion, this could be a summary of previous studies but not an article in its current form.

Response: Thank you very much for your comment. We have now improved the review based on your overall evaluation and according to the specific comments by reviewers 2 and 3. Starting from the results of the literature review and in line with the aim of the paper (please see our response to Comment 1b), we highlighted some findings that are reported now in the “Conclusion and future prospects section” as follows:

In order to make progress towards SDGs related to FLW reduction and to address the global food demand, the findings from this review suggest that the following four actions could be applied to the FSC: 1) to quantify how much food is lost and wasted, as well as where and why; 2) to explain and communicate the benefits of reducing FLW to the FSC actors; 3) to suggest strategies and tools for FLW reduction, optimizing the efficiency along the FSC; 4) to develop strategies to properly collect FBPW, recovering high value compounds, ensuring their traceability, safety and sustainability.

Moreover, in the same section, we added the sentence to further stress the possible strategies for preventing and reducing FLW:

Effective actions for preventing and reducing FLW have to involve all the main actors of the food chain that should share strategies for prevention, reduction and valorisation of food wastes. However, it is crucial to identify critical loss and waste points in the different supply chains as key steps for implementing effective countermeasures, which should be based on the peculiarities of the several production contexts.

Comment 1b: I couldn't even find the study's objectives, and the abstract exists in a different world than the rest of the manuscript.

Response: Thank for your comment. We better explained the aim of our paper both in the abstract and in the manuscript, also according to the comments from the others reviewers. Actually, our paper provides an overview rather than a framework, as better specified now in the sentence reported below:

In this challenging context, the aim of this paper is to provide an overview of the environmental, economic and social issues of FLW. This study may be helpful in identifying FLW causes and supporting the development of policies enhancing the sustainability of FSC.

Comment 1c: The authors aim to develop a framework for the environmental, economic, and social issues of FLW that may support strategies (as mentioned in the abstract), but the manuscript does not even discuss this framework in the later work. Therefore, with regret, I would request the editor to reject the manuscript this time.

Response: Thank you for your comment. Actually, our paper provides an overview rather than a framework, as better specified now in the sentence reported below and in the manuscript:

In this challenging context, the aim of this paper is to provide an overview of the environmental, economic and social issues of FLW. This study may be helpful in identifying FLW causes and supporting the development of policies enhancing the sustainability of FSC.

Therefore, in line with the aim of the paper and starting from the results of the literature review, we highlighted some findings that are reported now in the “Conclusion and future prospects section” as follows:

In order to make progress towards SDGs related to FLW reduction and to address the global food demand, the findings from this review suggest that the following four actions could be applied to the FSC: 1) to quantify how much food is lost and wasted, as well as where and why; 2) to explain and communicate the benefits of reducing FLW to the FSC actors; 3) to suggest strategies and tools for FLW reduction, optimizing the efficiency along the FSC; 4) to develop strategies to properly collect FBPW, recovering high value compounds, ensuring their traceability, safety and sustainability.

Finally, we deeply revised the abstract that is now more coherent with the aim and contents of the manuscript:

Abstract: To address global food security, new strategies are required in view of the challenges represented by Climate Change, depletion of natural resources and the need not to compromise further the ecosystems quality and biodiversity. Food Losses and Waste (FLW) affect food security and nutrition, as well as sustainability of food systems. Quantification of adverse effects of FLW is a complex and multidimensional challenge requiring a wide-ranging approach, regarding the quantification of FLW as well as the related economic, environmental and social aspects. The evaluation of suitable corrective actions for managing FLW along the food supply chain requires a system of sound and shared benchmarks that seem still undefined. This review aims to provide an overview of the environmental, economic and social issues of FLW, which may support policies and strategies for prevention, reduction and valorization of food wastes within the food supply chain. Detection of the hotspots and critical points allows to develop tailored policies, strategies and digital technologies for smart agriculture, improving the efficiency of the food supply chain and its sustainability, with an integrated approach involving all the main actors.

Reviewer 2 Report

1.      It is better to avoid the words used in the title for keywords, and replace the same with catchy keywords

2.      Rewrite the sentence “The food demand from the growing world’s population is significantly increasing the pressure on natural resources and ecosystems.” for better clarity.

3.      Why “Climate Change”  is in capital letter

4.      In the introduction section briefly describe the food requirement and global hunger status with suitable citations, the author can refer followings for the same-

1.      https://www.fao.org/3/cc0639en/cc0639en.pdf

2.      Jaiswal, D. K., Krishna, R., Chouhan, G. K., de Araujo Pereira, A. P., Ade, A. B., Prakash, S., ... & Verma, J. P. (2022). Bio-fortification of minerals in crops: current scenario and future prospects for sustainable agriculture and human health. Plant Growth Regulation, 1-18.

5.      Table 1 is ok

6.      Rewrite the sentence “The greatest amount of waste is generally created during transport and storage, washing, peeling, cutting, and cooking” for better clarity.

7.      Re-prepare figure 1 for better clarity and give a detailed caption below the figure, not above the figure.

8.      Table 2 is ok

9.      It is better to write the “present study or present review article” in place of “Current literature/paper etc.” so replace the same with a suitable one

10.  Rewrite the sentence “Moreover, evidences regarding adoption rates or the economic sustainability of FLW reduction actions are still scarce, and do not provide guidance on how to provide economic incentives for actors from farm-to-fork engaged in virtuous behaviors” for better clarity

11.  Table 3 is ok

12.  It is better to replace “Conclusive remarks” with “conclusion and future prospects”. For better soundness add a future roadmap to achieve the SDGs and to feed the growing global population by reducing food losses and waste. It will be also better to add a schematic flow diagram chart to reduce food losses and wastage.

Overall, the manuscript is presented in a nice but there are many minor grammatical mistakes and the use of unscientific words which can be improved. Globally, 10-24% population is suffering from hunger due to insufficient food. The present study may be helpful in identifying food losses and waste factors and the development of policies to achieve sustainable development goals. 

Author Response

Comment 1. It is better to avoid the words used in the title for keywords, and replace the same with catchy keywords.

Response: Thank you very much for this suggestion. As suggested by the reviewer 3, the title has been modified as: “Challenges for a sustainable food supply chain: A review on food losses and waste”. Moreover, we corrected some keywords.

Comment 2. Rewrite the sentence “The food demand from the growing world’s population is significantly increasing the pressure on natural resources and ecosystems.” for better clarity.

Response: Thank you very much for this remark.  The sentence has been rephrased as follows: “In the last decades, the pressure on natural resources and ecosystems has strongly increased with the need for food for the growing world’s population.”

Comment 3. Why “Climate Change” is in capital letter.

Response: Thank you for having given the chance to explain. Given the epoch-making impact of the phenomenon, as researchers fighting for the data-based science and against fake science, we feel that the use of capital letter would be a very little contribution to aware our politicians to implement actions to limit the Global Warming.

Comment 4. In the introduction section briefly describe the food requirement and global hunger status with suitable citations, the author can refer followings for the same: 1)       https://www.fao.org/3/cc0639en/cc0639en.pdf; 2) Jaiswal, D. K., Krishna, R., Chouhan, G. K., de Araujo Pereira, A. P., Ade, A. B., Prakash, S., ... & Verma, J. P. (2022). Bio-fortification of minerals in crops: current scenario and future prospects for sustainable agriculture and human health. Plant Growth Regulation, 1-18.

Response: Thank you very much for this suggestion.  We have added the following text at the beginning of the introduction section: “In the last decades, the pressure on natural resources and ecosystems has strongly increased with the need for food for the growing world’s population. Between 702 and 828 million people in the World faced hunger in 2021. Hunger affected 46 million more people in 2021 compared to 2020, and 150 million more people since 2019, before the Covid-19 pandemic. More than half of the people in the World affected by hunger in 2021 live in Asia and more than one-third in Africa (FAO, 2022)”.

In the introduction (page 2) we reported the following sentence: “Plant-based food is a key source of minerals and energy, providing almost 90% of the calories and 80% of protein requirements for daily human intake. Globally, about 210,000 M of hectares of agricultural land produces about 300,000 million of tons (Mt) of vegetables (Jaiswal et al., 2022)”.

Comment 5. Table 1 is ok.

Response: Thank you very much for this comment. 

Comment 6. Rewrite the sentence “The greatest amount of waste is generally created during transport and storage, washing, peeling, cutting, and cooking” for better clarity.

Response: Thank you very much for this remark. The sentence has been rephrased as follows: “The high amount of waste in processing is mainly due to the spoilage of commodities during transport and storage, and to the scrap arising from washing, peeling, cutting and cooking operations”.

Comment 7. Re-prepare figure 1 for better clarity and give a detailed caption below the figure, not above the figure.

Response: Thank you for this remark.  We have moved the caption below the figure, and rephrased the caption as follows: “Figure 1. Contribution of FLW (green columns) and ‘EU available’ food commodities (light blue columns). Modified from [1]”.

In the manuscript we revised the text as follow: “Caldeira et al. [2] estimated the contribution of FLW from each food group as well as the amount of ‘EU available’ food commodities (calculated as production, plus import, minus exports of primary commodities, minus non-food uses). Figure 1 shows that contribution on a plane of Cartesian axes wherein the horizontal axis reports the food groups, while the vertical axis reports the FLW and food commodity availability as percentage of their own total amount.”

Comment 8. Table 2 is ok.

Response: Thank you for this comment. 

Comment 9. It is better to write the “present study or present review article” in place of “Current literature/paper etc.” so replace the same with a suitable one.

Response: Thank you for this suggestion.  We intended that, in general, scientific literature devoted less attention to economic assessment of FLW in comparison to their quantification or environmental issues. We reformulated the sentence as follows: “At the best of our knowledge, scientific literature was more focused on FLW quantification, prevention, reduction and environmental impacts, and devoted less attention on the economic assessment of FLW and food by-product and wastes (FBPW).”.

Comment 10. Rewrite the sentence “Moreover, evidences regarding adoption rates or the economic sustainability of FLW reduction actions are still scarce, and do not provide guidance on how to provide economic incentives for actors from farm-to-fork engaged in virtuous behaviors” for better clarity.

Response: Thank you for this suggestion.  We have reformulated the sentence as follows: “Moreover, evidence about the efficacy of actions to reduce FLW and improve economic sustainability are still scarce, and do not guide key actors in obtaining economic incentives within a farm-to-fork virtuous approach”.

Comment 11. Table 3 is ok.

Response: Thank you for this comment. 

Comment 12. It is better to replace “Conclusive remarks” with “conclusion and future prospects”. For better soundness add a future roadmap to achieve the SDGs and to feed the growing global population by reducing food losses and waste. It will be also better to add a schematic flow diagram chart to reduce food losses and wastage.

Response: Thank you for this suggestion. We have replaced “Conclusive remarks” with “Conclusions and future prospects”. We have added the following text to specify the roadmap to achieve the SDGs and to feed the growing global population by reducing food losses and waste: “In order to make progress towards SDGs related to FLW reduction and to address the global food demand, the findings from this review suggest that the following four actions could be applied to the FSC: 1) to quantify how much food is lost and wasted, as well as where and why; 2) to explain and communicate the benefits of reducing FLW to the FSC actors ; 3) to suggest strategies and tools for FLW reduction, optimizing the efficiency along the FSC; 4) to develop strategies to properly collect FBPW, recovering high value compounds, ensuring their traceability, safety and sustainability.”.

Regarding the flowchart, we think that it can be not suitable for the “conclusion and future prospects” section, but we have added the following sentence to address your comment: “Effective actions for preventing and reducing FLW have to involve all the main actors of the food chain that should share strategies for prevention, reduction and valorisation of food wastes. However, it is crucial to identify critical loss and waste points in the different supply chains as key steps for implementing effective countermeasures, which should be based on the peculiarities of the several production contexts.”.

Comment 13. Overall, the manuscript is presented in a nice but there are many minor grammatical mistakes and the use of unscientific words which can be improved. Globally, 10-24% population is suffering from hunger due to insufficient food. The present study may be helpful in identifying food losses and waste factors and the development of policies to achieve sustainable development goals.

Response: Thank you for this remark.  We have added a description of the global hunger status according to comment 4, as well as the following sentence in the introduction: “This study may be helpful in identifying FLW causes and to support the development of policies enhancing the sustainability of FSC”. In addition, we carefully corrected the grammatical mistakes throughout the manuscript.

Reviewer 3 Report

Review report for sustainability-1995406-peer-review-v1:

Thank you very much for the opportunity. This study is conducted as a review research of food losses and waste concept. Kindly note the following issues which need to address before considering the manuscript for publication.

1. Title: The title is not interesting that should be modified, recommend the title as “Sustainable challenges in food supply chain: A review on food losses and waste”.

2. Abstract: 1st three lines must be punchy and represent the main issue. It looks very narrow and unable to justify the main issue, which needs to address that why this research is going so important? second, in abstract section do not use abbreviations.

3. The introduction section must start with the sustainable issue and global concern on the matter rather than discussing food waste, I mean need to address why food waste and losses is resource for high sustainable issues? Next, in introduction section the line
The High Level Panel of Experts on Food Security and Nutrition [2] defined Food losses (FL) as the “decrease, at all stages of the food chain prior to the consumer level, in mass, of food that was originally intended for human consumption, regardless of the cause” and Food waste (FW) as ”food appropriate for human consumption being discarded or left to spoil at consumer level – regardless of the cause”. Both food losses and waste (FLW) generation has dramatically increased over the last few decades
[3–5]
is unclear and need to rewrite. For example, use the term food waste without FW, same suggestion for food losses term. Over all introduction section is not clear to much need to update.

4. In the Quantification of FLW section, I am unable to find number of articles and data basis you have use to review the Food losses and waste concept. So, you must add this section and process of selection of articles. Further, in this section I find out again writing mistake such as
).” the supply chain (FSC)”, So, this article needs to improve too much.

5. Explain in detail about the interpretation of results in the discussion section with the help literature gaps.

6. Practical implications need to write separately, 

Author Response

Comment 1. The title is not interesting that should be modified, recommend the title as “Sustainable challenges in food supply chain: A review on food losses and waste.

Response: Thank you very much for this suggestion.  We have now changed the title as follows: “Challenges for a sustainable food supply chain: A review on food losses and waste”.

Comment 2. Abstract: 1st three lines must be punchy and represent the main issue. It looks very narrow and unable to justify the main issue, which needs to address that why this research is going so important? Second, in abstract section do not use abbreviations.

Response: Thank you very much for these remarks.  We added the following sentence at the beginning of the abstract to stress the main issue: “To address global food security and nutrition, new strategies are required in view of the challenges represented by Climate Change, depletion of natural resources and the need not to compromise further the ecosystems quality and biodiversity.”. We also limited the use of abbreviations (only one).

Comment 3. The introduction section must start with the sustainable issue and global concern on the matter rather than discussing food waste, I mean need to address why food waste and losses is resource for high sustainable issues?

Next, in introduction section the line “The High Level Panel of Experts on Food Security and Nutrition [2] defined Food losses (FL) as the “decrease, at all stages of the food chain prior to the consumer level, in mass, of food that was originally intended for human consumption, regardless of the cause” and Food waste (FW) as ”food appropriate for human consumption being discarded or left to spoil at consumer level – regardless of the cause”. Both food losses and waste (FLW) generation has dramatically increased over the last few decades [3–5]” is unclear and need to rewrite. For example, use the term food waste without FW, same suggestion for food losses term.

Over all introduction section is not clear to much need to update.

Response: Thank you very much for these remarks. We added the following sentence at the beginning of the introduction: “In the last decades, the pressure on natural resources and ecosystems has strongly increased with the need for food for the growing world’s population. Between 702 and 828 million people in the World faced hunger in 2021. Hunger affected 46 million more people in 2021 compared to 2020, and 150 million more people since 2019, before the Covid-19 pandemic. More than half of the people in the World affected by hunger in 2021 live in Asia and more than one-third in Africa (FAO, 2022)”.

We also deleted the abbreviations FL and FW from the definition by the High Level Panel of Experts on Food Security and Nutrition.

Comment 4. In the Quantification of FLW section, I am unable to find number of articles and data basis you have use to review the Food losses and waste concept. So, you must add this section and process of selection of articles.

Further, in this section I find out again writing mistake such as).” the supply chain (FSC)”, So, this article needs to improve too much.

Response: Thank you very much for these comments. This narrative literature review includes research articles and grey literature from reports by government and non-governmental organizations. Google scholar and Scopus databases were searched for the terms “food loss” AND/OR “food waste” AND/OR “by-product” AND/OR “environmental impact” AND/OR “environmental sustainability” AND/OR “social sustainability” AND/OR “economic sustainability”. Titles and abstracts of all studies detected by the search have been screened and selected according to a qualitative approach based on their ability to address the subject of food losses and waste, in relation to the three dimensions of sustainability. About a hundred studies have been included in the next phase, and after a deep reading ninety articles have been considered suitable for the review and grouped together according to the main. In particular, seventeen articles were considered relevant for food waste definition and classification aspects, twenty-four articles dealt with quantification of waste, twenty and twelve articles, respectively, contributed to the definition of economic and environmental impacts, and finally forty articles have been found effective in defining the social issues. We did not specify the above article selection process, because it is not usual for a narrative review, but it is mandatory for a systematic review.

We corrected all the mistakes throughout the manuscript, also in accordance with the comments by Reviewer 2.

Comment 5. Explain in detail about the interpretation of results in the discussion section with the help literature gaps.

Response: Thank you very much for this comment.  We improved the last section as follows, also in accordance with a remark by Reviewer 2: “In order to make progress towards SDGs related to FLW reduction and to address the global food demand, the findings from this review suggest that the following four actions could be applied to the FSC: 1) to quantify how much food is lost and wasted, as well as where and why; 2) to explain and communicate the benefits of reducing FLW to the FSC actors ; 3) to suggest strategies and tools for FLW reduction, optimizing the efficiency along the FSC; 4) to develop strategies to properly collect FBPW, recovering high value compounds, ensuring their traceability, safety and sustainability.”. “However, it is crucial to identify critical loss and waste points in the different supply chains as key steps for implementing effective countermeasures, which should be based on the peculiarities of the several production contexts.”.

Comment 6. Practical implications need to write separately.

Response: Thank you very much for this comment.  In the last paragraphs, we separated the practical implications from the rest of conclusions and future prospects.

Round 2

Reviewer 1 Report

I appreciate the effort of the authors who made significant changes in the revised version which inclined me to accept the article. 

Author Response

Response: Thank you for your valuable support for improving our work.

Reviewer 3 Report

Conclusion section:

There is a lack of clarity in the practical implications; might you respond to my queries by focusing on these? How can policymakers make use of this research? Exactly how can a government use this research to better ensure its own long-term sustainability?

Author Response

Response: Thank you for your comment. We better clarified the practical implications in the section “Conclusions and future prospects” as follows:

“In particular, the insights of this study may support the efforts of policy makers and government institutions, especially at local level, in defining policy measures that have to be designed and planned carefully considering the measurements of losses and waste, the supply chain where they occur and the critical loss points. Therefore, tailored policy measures should be structured at three different levels.

Firstly, measures aimed at preventing waste generation and ensuring food security have to be strengthened, in order to reduce inefficiencies in the food systems. For instance, these measures could improve: the management of edible and inedible parts of food at the primary production level, and the efficiency of storage, packaging and processing by adopting digital technologies for smart agriculture; the reduction of waste at retail level by the better management of ‘close-to-expiry’ products and food redistribution, as well as by stimulating the consumers’ awareness about food waste management through food and nutrition knowledge campaigns.

Secondly, specific policy measures could support the implementation of FW collection and treatment infrastructures and services, which should be designed in relation to the quantity and quality of food waste to be managed (e.g. recovery of high value added compounds, anaerobic digestion, composting, etc.). In this step, the quantification of environmental impacts, costs and benefits of FW reduction will be crucial to justify targeted financial resources’ allocation.

Finally, weighing the costs and benefits of FW reduction represents a guidance for designing policies in the view of the valorisation of waste as energy/material (e.g. using FW to produce high value compounds or biofuels). In this regard, the overview raised from this study may improve the awareness about the origin and composition of FW flows within the FSC, and to what extent these flows may vary over time. This may support the development of specific policy measures aimed to facilitate the implementation of innovative sustainable processes for bioactive molecules and functional ingredients recovery. These processes contribute to efficient management of waste, meanwhile taking advantage from an inexpensive source of valuable compounds with recognised positive effects on human health.

As practical implications, these possible policy measures must necessarily be based on integrated food chain approaches taking into account the strong link between FLW reduction and food security, within the horizontal and vertical cooperation among the FSC actors and institutions. Indeed, effective measures for preventing and reducing FLW have to involve all the main actors and institutions in sharing concrete strategies and tools. However, it is useful to identify critical loss and waste points along the different supply chains as key steps for implementing these measures, which should be based consequently on the peculiarities of the several production contexts.